# Current Perspectives on the Management of Herpesvirus Infections in Solid Organ Transplant Recipients

**DOI:** 10.3390/v15071595

**Published:** 2023-07-21

**Authors:** S. Reshwan K. Malahe, Jeroen J. A. van Kampen, Olivier C. Manintveld, Rogier A. S. Hoek, Caroline M. den Hoed, Carla C. Baan, Marcia M. L. Kho, Georges M. G. M. Verjans

**Affiliations:** 1Department of Internal Medicine, Erasmus University Medical Center, 3015 GD Rotterdam, The Netherlands; s.malahe@erasmusmc.nl (S.R.K.M.); c.c.baan@erasmusmc.nl (C.C.B.); m.kho@erasmusmc.nl (M.M.L.K.); 2Erasmus MC Transplant Institute, Erasmus University Medical Center, 3015 GD Rotterdam, The Netherlands; o.manintveld@erasmusmc.nl (O.C.M.); r.hoek@erasmusmc.nl (R.A.S.H.); c.denhoed@erasmusmc.nl (C.M.d.H.); 3Department of Viroscience, Erasmus University Medical Center, 3015 GD Rotterdam, The Netherlands; j.vankampen@erasmusmc.nl; 4Department of Cardiology, Erasmus University Medical Center, 3015 GD Rotterdam, The Netherlands; 5Department of Pulmonary Medicine, Erasmus University Medical Center, 3015 GD Rotterdam, The Netherlands; 6Department of Gastroenterology and Hepatology, Erasmus University Medical Center, 3015 GD Rotterdam, The Netherlands; 7HerpeslabNL, Department of Viroscience, Erasmus University Medical Center, 3015 GD Rotterdam, The Netherlands

**Keywords:** human herpesviruses, solid organ transplantation, solid organ transplant recipients

## Abstract

Solid organ transplant recipients (SOTRs) are at high risk of human herpesvirus (HHV)-related morbidity and mortality due to the use of immunosuppressive therapy. We aim to increase awareness and understanding of HHV disease burden in SOTRs by providing an overview of current prevention and management strategies as described in the literature and guidelines. We discuss challenges in both prevention and treatment as well as future perspectives.

## 1. Introduction

Solid organ transplantation (SOT) has transformed the survival and quality of life of patients with end-organ dysfunction. It has become the standard of care for approximately a hundred thousand patients worldwide each year, and offers long-term treatment to patients with otherwise limited life expectancy [1]. Decades of improvement on prevention and therapeutic interventions, including surgical techniques and stronger immunosuppressive regimens, increased the quality of life of people suffering from debilitating organ diseases and led to a better graft survival in SOTRs [2]. However, these advances come with a price as immunosuppression increases the risk of morbidity and mortality from infections [3]. Herpesvirus infections remain a major cause of morbidity and mortality among SOT recipients (SOTRs) [4]. Herpesviruses are large double-stranded DNA viruses that infect members of all groups of vertebrates. The long-term co-speciation of the ancestral herpesvirus and its various hosts has resulted in virus’s adaptation and consequently extraordinary species’ specificity and intra-species divergence, resulting in more than 150 different herpesviruses being identified to date [5]. Humans are a natural host of nine distinct herpesviruses, human herpesviruses (HHV)1–8, (HHV6 species are divided into HHV6A and HHV6B), each causing a characteristic disease in humans due to their route of transmission, cell tropism, and replication kinetics [6]. The hallmark of herpesviruses is their ability to establish a lifelong latent, nonproductive infection of the host, which can be reactivated at later times to cause associated disease. Reactivation can occur when immunity wanes due to older age, underlying disease, or immunosuppressive therapy [3,4]. Current preventive measures of HHV-induced diseases include determining the serostatus in the donor and recipient, viral load monitoring, antiviral prophylaxis, preemptive therapy, vaccination, immunosuppression adjustment, and education and hygiene. Antiviral prophylaxis involves administering antiviral drugs either to all patients (universal prophylaxis) or selectively to a subgroup of patients at higher risk of HHV infection (specific prophylaxis). Preemptive therapy involves closely monitoring patients for productive HHV infection and administering antiviral drugs only when viral replication is detected, aiming to prevent progression to disease [7]. Therapeutic strategies include antiviral treatment, immunosuppression adjustment, and the management of complications. In this review, we present incidence rates, complications, and recent advances in preventive measures and the management of HHV infections in SOTRs. The order of HHVs discussed is based on the prevalence of their post-SOT complications.

## 2. Human Cytomegalovirus

### 2.1. Incidence

Human cytomegalovirus (HCMV; HHV5) infections are most prevalent among solid organ transplant recipients (SOTRs), causing severe morbidity and mortality [8,9]. HCMV seroprevalence depends on geography, age, and socioeconomic status [10]. The global HCMV seroprevalence is estimated to be 66% in Europe, 75% in the Americas, 86% in south-east Asia, 88% in the African and the western Pacific regions, and 90% in the eastern Mediterranean region [11].

### 2.2. Complications

The most updated definition of HCMV infection is the isolation of the virus or the detection of viral proteins or DNA in any body fluid or tissue specimen. HCMV-induced disease in SOTR consists in “end-organ disease” and “CMV syndrome”. A proven HCMV end-organ disease is defined as having clinical symptoms and/or signs combined with the detection of HCMV in tissue from the affected organ. The CMV syndrome definition requires the detection of HCMV in blood by viral isolation, rapid culture, antigenemia, or PCR analysis, combined with at least two of the following parameters: fever, malaise, leukopenia/neutropenia, atypical lymphocytes, thrombocytopenia, and the elevation of hepatic aminotransferases. The CMV syndrome is the most common presentation of HCMV-induced disease in SOTRs [12]. HCMV infections of the allograft are more often observed in the liver (LiTR), lung (LuTR), and vasculopathy in heart transplant recipients (HTRs) [13]. The transplantation of an organ from an HCMV seropositive donor to a HCMV seronegative recipient, a so-called donor (D)+/recipient (R)− combination, poses the highest risk of primary infection. HCMV infection is associated with a higher risk of graft loss and mortality in SOTRs [13,14,15,16]. This association cannot be easily explained. Multiple infection-related factors influence graft loss and mortality: end-organ disease, indirect effects such as increased severity of atherosclerosis, and altered immune responses due to patient characteristics, immunosuppressive medication, and HCMV itself [17].

### 2.3. Prevention

#### 2.3.1. Parameters for Risk Stratification

Determining the HCMV serostatus of transplant candidates and donors is pivotal to determining the risk of post-SOT HCMV-induced diseases [7,9]. Hypogammaglobinemia commonly occurs in SOTRs due to immunosuppressive therapy and is an independent risk factor for the development of HCMV infection post-transplantation [18,19,20]. There are other parameters that can be used to estimate the risk of infection including low lymphocyte counts, complement levels, and natural killer cell counts, but given the complexity of the immune response, it is not likely that a single marker will be superior in risk prediction [9].

#### 2.3.2. Strategies for Preventing HCMV Disease

Antiviral prophylaxis and preemptive therapy are the two main strategies to prevent HCMV disease [7,9]. Both strategies have their advantages and disadvantages. The advantages of antiviral prophylaxis over preemptive therapy are that its clinical efficacy is based on large randomized controlled trials, so that it is easier to coordinate and has lower laboratory costs and positive indirect effects (on graft loss, mortality, and opportunistic infections). The advantages of preemptive therapy over antiviral prophylaxis are that it is associated with less delayed-onset HCMV disease due to HCMV immune reconstitution, has lower drug costs and lesser drug toxicity with shorter courses of antiviral therapy [7,9]. These factors must therefore be taken into account by the individual center when choosing the prevention method [9]. For D+/R− and R+, both antiviral prophylaxis and preemptive therapy guided by HCMV DNAemia in plasma have shown comparable efficacy in KTR and LiTR, while only antiviral prophylaxis is well studied in LuTR and HTR [7,9]. However, two recent randomized trials in D+/R− LiTR reported that preemptive therapy was more efficacious in preventing symptomatic HCMV disease and more cost-effective compared to antiviral prophylaxis [21,22]. In addition, in LiTR patients, preemptive therapy was associated with the induction of both higher HCMV-specific CD8 T-cell responses and neutralizing antibodies compared to antiviral prophylaxis, which could result in better protection against late-onset disease [23]. Monitoring as part of preemptive therapy is recommended at least once weekly for 3–4 months after kidney or liver transplantation and may be extended if there is an ongoing increased risk for HCMV disease [9]. Although there is no consensus for the optimal HMCV-DNAemia cut-off for initiating preemptive therapy, a recent retrospective study showed that using a threshold of ≥4000 IU/mL was effective in reducing the incidence of end-organ disease in R+ LiTR [24]. Thresholds for initiating preemptive therapy are usually defined on available assays and patient risk factors [9]. Besides transmission during an organ transplant, HCMV could also be transmitted through blood transfusion, and therefore it is recommended to use leukocyte-reduced or HCMV-seronegative blood products in high-risk patients [9]. Also, there has recently been more interest in the research of monoclonal antibodies to prevent infection. A phase 2 trial showed that a combination of two anti-HCMV monoclonal antibodies was associated with less HCMV DNAemia and disease, and was well tolerated compared with the placebo [25]. Since hypogammaglobinemia is an independent risk factor for the development of HCMV infection post-transplantation, it is conceivable that intravenous immunoglobulins (IVIGs), a pool of antibodies from the plasma of healthy donors, may prevent HCMV disease. However, there is currently insufficient data to support the use of IVIG as prophylaxis to reduce infection in SOTRs [18,19,20,26]. Furthermore some studies suggest that antivirals combined with IVIG may be superior in SOTRs, but these findings are still under debate [27,28].

#### 2.3.3. Nucleoside Analogues

Valganciclovir is currently the first choice for prophylaxis. The recommended valganciclovir dose is 900 mg/day, with dose adjustments in patients with renal impairment [7,9]. The importance of accurate dose adjustment to the most recent estimated glomerular filtration rate (eGFR) value is emphasized by data showing that SOTRs that received doses below the eGFR-adjusted dosages recommended by the manufacturer were at an increased risk of HCMV breakthrough infection and viral resistance [29]. Some transplant centers use 450 mg/day to decrease the risk of leucopenia, but lower drug exposition is also associated with the emergence of drug-resistant HCMV, particularly in D+/R− recipients, and is therefore not advocated [7,9,30]. Pharmacokinetic simulations showed that 79% of recipients could still not attain therapeutic targets despite 100% matched guideline valganciclovir prophylaxis dosing, while 6% (3/50) developed HCMV breakthrough infection and 12% (6/50) developed late-onset infection. Therefore, it may be useful to perform ganciclovir therapeutic drug monitoring and the determination of the daily area under the curve (AUC24) in high-risk patients [31]. The recommended duration of prophylaxis for KTR is 6 months for D+/R− and 3 months for R+ combinations; for LiTR and HTR, it is 3–6 months for D+/R− combination and 3 months for R+ combinations; and for LuTR, it is 6–12 months for all combinations [7,9]. However, the incidence of primary infection after the cessation of antiviral prophylaxis remains high (38%) in D+/R− KTR despite the administration of the recommended 6 months post-transplantation and surveillance after prophylaxis. Surveillance after the cessation of prophylaxis may be considered in high-risk patients, although evidence from research is low [32]. For D−/R− SOTRs, routine HCMV prevention is not recommended [7,9]. After the treatment of rejection with lymphocyte-depleting anti-lymphocyte antibodies (e.g., anti-thymocyte globulin or alemtuzumab), SOTRs should also receive prophylaxis (with valganciclovir) [7,9]. However, little is known about the optimal duration of prophylaxis in this case.

#### 2.3.4. Non-Nucleoside Analogues

Letermovir is a non-nucleoside viral terminase complex inhibitor, about 1000 times more effective than ganciclovir against HCMV in vitro, available for oral and intravenous use and has limited side effects [33]. In addition to being approved by the FDA and European Medicines Agency (EMA) as HCMV prophylaxis in adult HCMV-seropositive HSCT recipients, letermovir has also recently been approved by the FDA for the prevention of HCMV disease in D+/R− adult KTR [34]. This decision was supported by a recent phase 3 randomized double-blind trial demonstrating that letermovir was non-inferior to valganciclovir in preventing HCMV disease in 601 D+/R− (high-risk) KTR [35]. For the other SOTRs, letermovir has also shown efficacy in a recent clinical trial to prevent HCMV infection and disease in adult LuTR, and in a small study with high-risk HTR who could not tolerate valganciclovir [36,37]. However, there are already a few cases of letermovir resistance in SOTRs [38,39,40]. In addition, case reports showed the mixed efficacy of letermovir in LuTR with refractory or resistant HCMV infection [41]. Also, drug–drug interactions between letermovir and immunosuppressants including cyclosporine, tacrolimus, and sirolimus have been described [42,43].

#### 2.3.5. Cell-Mediated Immunity Assays

Another method to predict the risk of the development of HCMV disease post-transplantation could be the use of other biomarkers. A promising development is the use of in vitro cell-mediated immune assays in the blood of SOTRs for the prevention and management of HCMV disease. As an example, determining HCMV-specific T-cell immunity by interferon-ɣ release assays, such as the QuantiFERON-CMV assay, is of additive value to determine the duration of antiviral prophylaxis compared with a fixed duration, as shown in LuTR within 18 months post-transplantation [44]. Other examples are the use of the ELISpot-HCMV assay and intracellular cytokine staining to determine the individual’s virus-specific T-cell immune status [45,46]. Further research is needed to determine whether these T-cell assays are of additional prognostic value in high-risk (D+/R−) recipients [47]. Furthermore, these assays should be standardized and cut-off values for ELISpot and intracellular cytokine staining assays are yet to be defined [9]. Additional applications of assessing HCMV-specific immunity are using ELISpot assay to predict protection from HCMV infection; determining pre-transplant interferon-ɣ response to identify R+ recipients that can clear HCMV reactivation post-transplantation without antiviral treatment, to identify R+ recipients who are at higher risk of developing HCMV reactivation, to identify R+ recipients who can discontinue prophylaxis early after antithymocyte globulin induction therapy, and to predict who will have a high level of viral replication after prophylaxis withdrawal [48,49,50,51,52]. A dysfunctional T-cell phenotype could also be a new biomarker. Dysfunctional γδ and αβ T-cells have been observed in R+ KTR who were treated with mycophenolate mofetil. This dysfunctional T-cell profile was associated with severe HCMV infection, while mTOR inhibitors were shown to enhance the proportion of functional T-cells. Therefore, a dysfunctional T-cell profile could be used as a biomarker to predict the post-transplantation infection and indicate who should benefit from treatment with mTOR inhibitors [53].

### 2.4. Treatment

Oral valganciclovir is recommended as first-line treatment in the case of mild-to-moderate HCMV disease in patients who can tolerate and adhere to oral medication. Intravenous ganciclovir is recommended in the case of severe HCMV disease or when there are concerns about absorption [7,9]. If DNAemia has reached a positive threshold during monitoring in the context of preemptive therapy, valganciclovir should be started, with a minimum of 2 weeks, until the resolution of DNAemia. After treatment, weekly surveillance should be started [9]. For symptomatic patients, antiviral treatment should be continued for at least 2 weeks, until the disease is clinically resolved and HCMV DNAemia is eradicated in one or two consecutive weekly samples. During treatment, HCMV DNA should be tested in plasma weekly to monitor response and renal function should be frequently monitored to guide dose adjustments [9]. It might be a challenge to determine the optimal dose of valganciclovir in patients with impaired kidney function. Serum ganciclovir trough level monitoring is not recommended in guidelines as it has not been significantly associated with improved clinical outcome. However, it could be used to determine the optimal dose in patients with impaired kidney function and in children [7,9]. A recent population pharmacokinetic/dynamic model of valganciclovir for preemptive therapy showed that a dose increase to 450 mg every 36 h may reduce the time to optimal viral load target in patients with an eGFR of 10–24 mL/min/1.73 m^2^ for the thresholds of ≤290 and ≤137 IU/mL [54]. Patients who received (val)ganciclovir for more than 6 weeks in the past and have not responded to at least 2 weeks of antiviral treatment or have developed HCMV DNAemia during prophylaxis, should be suspected for drug resistance [9]. In the case of refractory disease due to ganciclovir resistance, the recommended strategy is to cautiously reduce immunosuppressive therapy and switch to an unrelated antiviral like foscarnet or cidofovir, but their nephrotoxicity may be a major disadvantage [7,9]. An alternative option is high-dose intravenous ganciclovir [7,9]. However, this could lead to more cross-resistance to cidofovir and foscarnet, since continuing ganciclovir could eventually result in mutations in the viral *UL54* gene. Myelotoxicity, especially leucopenia, is another reason to discontinue (val)ganciclovir treatment. Discontinuing other myelosuppressive therapies and/or adding a granulocyte colony-stimulating factor should be considered first before switching to another agent [9]. The mammalian target of rapamycin (mTOR) inhibitors is associated with a reduced risk of HCMV infection/disease in R+ recipients, which has also recently been shown in D+/R− recipients [9,55]. Therefore, switching to an mTOR inhibitor could be the part of the management of (antiviral resistant) HCMV disease in SOTR [56,57,58]. The regular monitoring of HCMV-specific systemic T-cell immunity is proposed to be of additive value in the clinical management of HCMV complications post-SOT [59]. Indeed, a recent interventional study showed, for the first time, that the real-time measurement of systemic HCMV-specific T-cell immunity was a safe and feasible strategy for the early discontinuation of antiviral treatment [60].

Recently, maribavir was approved by the Food and Drug Administration (FDA) and the European Medicines Agency (EMA) for the treatment of refractory HCMV infection/disease in transplant patients [61]. Maribavir is a benzimidazole inhibitor with high anti-HCMV activity and favorable properties during (pre)clinical testing. In a recent phase 3 randomized clinical trial in patients with refractory disease, maribavir was superior to standard therapy (valganciclovir/foscarnet/cidofovir) in clearing HCMV DNAemia, symptom control, and was associated with less acute kidney injury compared to foscarnet and less neutropenia versus (val)ganciclovir [62]. In addition, it may also be an option for the first-line preemptive treatment of HCMV DNAemia [63]. However, the use of maribavir also has a number of disadvantages. Most importantly, maribavir has a poor penetration into the central nervous system and should therefore not be used to treat the HCMV infection of this compartment including the retina [64]. Furthermore, maribavir cannot be co-administered with ganciclovir because maribavir antagonizes the antiviral action of ganciclovir by inhibiting the viral *UL97*-mediated phosphorylation [65]. In addition, some mutations in *UL97* result in cross-resistance between ganciclovir and maribavir, which deprives the use of maribavir in the case of ganciclovir-resistant HCMV infection [66,67]. Moreover, maribavir is only available as an oral formulation [68]. In clinical practice, the approach to HMCV management appears to be heterogeneous across hospitals and countries despite international guidelines [69].

## 3. Epstein–Barr Virus

### 3.1. Incidence

More than 90% of the adult population globally is infected with the Epstein–Barr virus (EBV; HHV4) [70]. Its seroprevalence varies by geographic location and primary infection occurs at a younger age among persons of lower socioeconomic status [71]. Donor transmitted EBV infection is very common in D+/R− recipients [72].

### 3.2. Complications

EBV-induced post-transplant lymphoproliferative disease (PTLD) is a life-threatening complication. The risk of lymphoproliferative disease is 12-fold higher in KTR compared with a matched non-transplant population [73]. PTLD is associated with EBV in >80% cases in Europe and United States [74,75]. The negative EBV serostatus of the recipient is the greatest risk factor of EBV-induced PTLD, since 90% of early onset PTLD occurred in seronegative recipients and mostly in the first year post-transplantation [76]. The 5 years of cumulative incidence of PTLD has been reported to be 2.8% in lung, 1.7% in kidney, 0.9% in liver, and 0.7% in heart recipients. The cumulative incidence in KTR is equal to LuTR at 10 years post-transplantation [77]. EBV-seronegative KTR receiving polyclonal anti-lymphocyte antibodies or immunosuppression with belatacept (a co-stimulation blocker) are at high risk of PTLD [78,79,80]. Induction therapy with rituximab was associated with a reduced risk of PTLD [81]. Besides PTLD, EBV disease can manifest as infectious mononucleosis, as well as organ-specific (e.g., hepatitis, gastro-intestinal symptoms) and hematological complications. Examples of more severe hematologic complications are macrophage activation syndrome and hemophagocytic lymphohistiocytosis [82].

### 3.3. Prevention

Knowledge of the EBV serostatus in transplant candidates and donors is essential to determine the risk of EBV-induced disease post-transplantation [82]. Mostly, anti-VCA and anti-EBNA-1 IgG are determined [83]. The periodic monitoring of EBV DNAemia in the blood of D+/R− recipients during the first year after transplantation is used as a predictor for PTLD. When the viral threshold is reached, immunosuppressive treatment could be reduced to prevent PTLD onset. It also enables the early detection and preemptive treatment of EBV-induced diseases [84,85]. When primary EBV infection is detected in blood, both a reduction in immunosuppressive therapy and preemptive intervention are recommended [82]. The use of IVIGs, antivirals, and immunotherapy for the prevention of PTLD in EBV-mismatched recipients are not advocated [82]. In EBV-seropositive transplant recipients, viral load surveillance and preemptive strategies are also not recommended [82]. Although valganciclovir and (val)acyclovir show a partial effect against EBV, there is no evidence that antiviral prophylaxis or preemptive treatment with these antivirals prevents early and late EBV-induced PTLD in SOTRs [81,86].

### 3.4. Treatment

The first-line treatment is to reduce immunosuppressive therapy for nearly all early and late B-cell PTLD which are not rapidly progressive. However, this strategy does not apply to Burkitt and Hodgkin lymphoma. Depending on the type of lymphoma, rituximab (a depleting anti-CD20 antibody) and/or cytotoxic chemotherapy (R-CHOP 21) are recommended. This treatment requires hematological expertise. The use of antivirals with(out) IVIG is neither recommended as monotherapy, nor as adjunctive therapy [82].

## 4. Varicella Zoster Virus

### 4.1. Incidence

Varicella zoster virus (VZV; HHV3) seroprevalence is dependent on demographic, climatic and socioeconomic factors [87]. For example, the overall VZV seroprevalence in the Caribbean Netherlands is considerably lower than in the Netherlands (78% versus 95%, respectively) [88,89].

### 4.2. Complications

Herpes zoster (HZ), caused by the reactivation of latent VZV, is a frequent complication among SOTR and its incidence depends on seroprevalence and type of organ transplanted. Indeed, the incidence (cases/1000 person years) has been reported to be 55 in lung, 32 in heart, 20 in kidney, and 18 in LiTR [90,91,92,93]. Risk factors for HZ include: older age (≥50 years at transplantation), being an HTR or LuTR, lack of HCMV prophylaxis, and the use of mycophenolate mofetil in HTR and LiTR [92,93,94,95]. Primary VZV infection (varicella) is rare but can lead to severe complications and even death in SOTRs. The presentation of both HZ and varicella can be atypical [96].

### 4.3. Prevention

Strategies to prevent VZV-induced complications include: the vaccination of both VZV-seronegative and -positive transplant candidates to prevent varicella and HZ, respectively, as well as the ring vaccination of household members to reduce VZV transmission and antiviral prophylaxis [96]. VZV-seronegative transplant candidates should be vaccinated with two doses of a live-attenuated varicella vaccine pre-transplantation [97]. This is supported by a study that showed that varicella vaccination resulted in the seroconversion and induction of VZV-specific T-cell immunity in VZV-seronegative kidney transplant candidates [98]. SOTRs who remain VZV-seronegative are eligible for post-exposure prophylaxis (PEP) with a VZV immunoglobulin preparation (e.g., VariZIG^®^) within 10 days after exposure [96,97]. For the prevention of HZ, there are currently two approved and commercially available vaccines: a live-attenuated virus vaccine and a recombinant subunit vaccine [99]. It is recommended to vaccinate VZV-seropositive transplant candidates ≥50 years of age with an HZ vaccine. Preference is given to the recombinant subunit vaccine as the live-attenuated virus vaccine (however effective) should be administered no less than 4 weeks prior to transplant and could therefore delay organ transplantation [97,100]. Live-attenuated virus vaccines are contra-indicated in patients on immunosuppressive medication [101]. Notably, the recombinant subunit vaccine is superior in terms of the safety, efficacy, and durability of protection against HZ in healthy adults, including the elderly [102]. Also, a recent trial showed a similar profile in KTR [103].

Short-term valacyclovir prophylaxis is only recommended for VZV-seropositive SOTR who are not receiving HCMV prophylaxis, since HCMV prophylaxis (except letermovir) is also effective against VZV [96,104]. The use of antivirals as PEP has not yet been evaluated in randomized trials in immunocompromised individuals.

### 4.4. Treatment

Recommended treatments for post-SOT VZV diseases are intravenous acyclovir for primary varicella and severe HZ disease (e.g., disseminated HZ and central nerve involvement) and oral valacyclovir or famciclovir for localized HZ [96]. However, in immunocompromised cancer patients, oral valacyclovir (1000 mg three times a day) has been shown to provide comparable protection compared to intravenous acyclovir (5 mg/kg three times a day) [105]. Therefore, based on pharmacokinetic data, it may be permissible to treat all forms of HZ with oral valacyclovir in patients who can tolerate and adhere to oral medication, which might be more convenient and more cost-effective. In case of allergy or resistance to these antivirals, foscarnet and cidofovir are alternatives [96]. Immunoglobulin therapy, either non-specific or VZV-specific, is not recommended as treatment and has only been used anecdotally in a few cases [106,107,108,109,110].

## 5. Herpes Simplex Virus

### 5.1. Incidence

The seroprevalence of herpes simplex virus types 1 (HSV-1; HHV1) and 2 (HSV-2; HHV2) largely depends on geography, age, and socioeconomic factors. Globally, about 67% and 13% of individuals <50 years old are infected with HSV-1 and HSV-2, respectively [111]. The incidence of HSV infection in SOTRs has been reported to be 29 cases/1000 person years in Switzerland [104].

### 5.2. Complications

Clinical HSV manifestations, due to primary infection or reactivation, are mainly mucocutaneous lesions [112]. Compared with immunocompetent individuals, SOTRs show more frequent and higher HSV mucosal shedding, more severe clinical manifestations (e.g., visceral and central nervous system infections) and lower response to therapy [113,114]. Notably, HTR and LuTR have the highest risk of developing pneumonitis compared with other SOTRs [115]. Recently, a rare case of hemorrhagic allograft nephritis due to HSV-1 was described in a KTR who did not receive HCMV prophylaxis at day 10 after transplantation [116].

### 5.3. Prevention

Universal donor screening is not recommended in countries with a high HSV seroprevalence [117]. Antiviral prophylaxis is only recommended for HSV seropositive transplant recipients who are not receiving HCMV prophylaxis, since HCMV prophylaxis (except letermovir) is also effective to control HSV infections [104,112]. In addition, oral antiviral prophylaxis may be considered in KTR with frequent recurrent HSV infection [118]. Furthermore, behavioral methods are recommended to prevent HSV acquisition such as avoiding contact with persons with active lesions [112]. A large retrospective analysis showed no association between various maintenance immunosuppressive regimens or induction therapy and the risk for symptomatic HSV reactivation [104].

### 5.4. Treatment

Limited mucocutaneous HSV lesions can be treated with valacyclovir, while intravenous acyclovir is recommended for severe clinical manifestations [112]. In the case of HSV resistance to (val)acyclovir, foscarnet or cidofovir can be used as alternative treatments [112].

## 6. Human Herpesvirus 8

### 6.1. Incidence

Human herpesvirus 8 (HHV8) seroprevalence has a significant geographic diversity: 3–7% in the United States, 5% in North Europe, 5–20% in the Mediterranean, and >50% in sub-Saharan Africa [119,120,121,122]. This correlates with the inter-regional incidence of post-transplant HHV8-induced Kaposi’s sarcoma (KS). The world-wide incidence (cases/100,000 person years) of KS has been reported to be 96 in kidney, 49 in heart, 44 in liver, and 11 in lung recipients [123].

### 6.2. Complications

HHV8-induced disease, diagnosed as DNAemia or clinically manifesting as KS, is commonly due to reactivation in recipients who were seropositive before the transplant (R+) [124,125,126]. It can also be due to primary infection by graft-to-host transmission, HHV8-transformed cells present within the graft tissue or de novo infections [127,128]. The risk of KS is low in SOTR, but still between 200- and 500-fold greater compared with the general population [129,130]. Risk factors for KS include: male sex, non-Caucasian ethnicity, and older age upon transplantation. After the first year of transplantation, the incidence appears to decrease [131]. This can be attributed to the pivotal role of T-cells in controlling HHV8 infection, which implicates that the intensity of immunosuppressive treatment and the use of antilymphocyte agents increase the risk of HHV8-induced disease following transplantation. However, no association was found between KS and a specific induction or maintenance immunosuppressive therapy [131].

Besides KS, HHV8 may incidentally cause primary effusion lymphoma, forms of multicentric Castleman disease and severe, non-neoplastic complications such as hemophagocytic lymphohistiocytosis [132,133]. Recently, a syndrome called ‘’Kaposi sarcoma inflammatory cytokine syndrome’’ has been observed in an HIV-negative SOTR [134]. Its disease characteristics include: unexplained fever, severe systemic inflammation, and the elevated HHV8 viral load in blood. These symptoms closely resemble the first cases of this syndrome in HIV-infected patients. Although this relatively new syndrome clinically resembles multicentric Castleman disease, it is a different entity that warrants attention.

### 6.3. Prevention

Plasma HHV8 viral load monitoring is recommended for seronegative recipients or mismatched recipients in endemic countries [124,135,136]. The reduction in immunosuppressive therapy is recommended in cases of primary HHV8 infection or reactivation [137]. In high-risk patients with HHV8 DNAemia, the conversion to an mTOR inhibitor might be helpful [138,139,140,141,142].

### 6.4. Treatment

In the case of HHV8-induced disease post-SOT, first-line treatment is to cautiously reduce immunosuppressive therapy and, if applicable, convert calcineurin inhibitors into mTOR inhibitors [137]. This is further supported by recent in vitro experiments that showed that sirolimus promotes the autophagy-mediated cell death of Kaposi’s sarcoma cells [143]. If KS is unresponsive to the aforementioned strategies, patients may benefit from chemotherapy [137,142].

## 7. Human Herpesvirus 6 and 7

### 7.1. Incidence

Most infections with human herpesvirus 6A (HHV6A), human herpes virus 6B (HHV6B), and human herpes virus 7 (HHV7) are due to reactivation in SOTRs, since most adults (90%) are seropositive for both HHV6 and HHV7 [144,145]. The prevalence of HHV6 reactivation varies from 20 to 82% and that of HHV7 from 0 to 46% in SOTRs, and occurs most frequently within the first 2–4 weeks post-transplantation [146,147,148,149,150,151,152,153].

### 7.2. Complications

The majority of reactivations are asymptomatic, but some can manifest as fever, rash, bone marrow suppression, and organ involvement such as hepatitis, colitis, pneumonitis, and, sporadically, encephalitis [149,150,154]. Regarding the impact on the allograft, only high-grade HHV6 DNAemia (>10^5^ copies/mL) is associated with acute allograft rejection in HCMV D+/R− LiTR [155].

### 7.3. Prevention

Preventive measures such as antiviral prophylaxis, viral load monitoring, and preemptive therapy are not recommended as most infections are subclinical and transient. Moreover, there is insufficient evidence on the effectiveness for the screening for HHV6B viremia, antiviral prophylaxis (foscarnet), and preemptive therapy (foscarnet/ganciclovir) to prevent HHV6 encephalitis in HSCT recipients [137]. In addition, given the toxicity of the currently available antivirals, preventive measures should not be routinely performed [156].

### 7.4. Treatment

In cases of moderate-to-severe disease, antiviral therapy could be provided in combination with a reduction in immunosuppressive therapy [137]. Foscarnet was recently approved as the first-line treatment for HHV6 encephalitis in HSCT recipients in Japan [156,157,158]. However, this is the only licensed antiviral for HHV-6 and there are currently no licensed antivirals for HHV7 [137,156]. In addition, most recommendations for antiviral therapy is based on in vitro data only or based on clinical studies performed in HSCT recipients [156,159,160,161]. These data suggest that ganciclovir, foscarnet, and cidofovir are all able to inhibit HHV6, while only foscarnet and cidofovir are able to inhibit HHV7. Cidofovir and brincidofovir were shown to have the lowest half-maximal effective concentration values for HHV6B in vitro [162]. All these antivirals may be used in the case of HHV6 and HVV7 disease in SOTRs, taking into account the clinical context and potential drug toxicities [137,156].

## 8. Future Perspectives

Due to the viruses’ high prevalences worldwide, the majority of adults are latently infected with at least five of nine human herpesviruses. HHV infections remain a major cause of disease in SOTRs due to primary infection or reactivation [4,5]. These challenges call upon the development of novel antivirals, vaccines, adoptive immunotherapy, and ultimately therapies that inactivate latent HHV to prevent reactivation in SOTRs. Due to the great variation in HHV genetics and replication cycle, the development of HHV-wide interventions are utopic.

### 8.1. Antivirals

The registered first-line antivirals currently available are the nucleoside analogues famciclovir, (val)acyclovir, and especially (val)ganciclovir, which are limited to control HCMV, VZV, and HSV infections. Compared with acyclovir, ganciclovir has a broader activity by inhibiting the replication of HHV6 and HHV8, and is more potent against HCMV [154,163,164]. However, the emergence of antiviral resistance, reported in 3–30% depending on SOT and HHV, and their toxicity profile (e.g., hemato- and nephrotoxicity) in long-term use are major problems [68]. Toxicity also limits the use of second line antivirals such as foscarnet and cidofovir, despite their broad spectrum of antiviral activity to multiple HHVs. For example, a recent retrospective analysis of 16 SOTRs with ganciclovir-resistant/refractory HCMV infection treated with cidofovir showed that 37.5% developed nephrotoxicity and 25% uveitis, while the clearance of DNAemia occurred in only 50% [165]. This highlights the need for the development of less toxic antivirals.

Several new antivirals have shown promising (pre)clinical results including amenamevir and pritelivir [68]. Amenamevir and pritelivir are helicase-primase inhibitors, showing positive results in treating VZV and HSV infections, are well tolerated, and are superior to acyclovir and ganciclovir [166,167]. Oral pritelivir has shown effectivity in treating HSCT recipients with recrudescent acyclovir-resistant HSV-2 genital herpes [168]. Amenamevir shows a synergistic effect with acyclovir and ganciclovir towards both VZV and HSV replication in vitro, which may be of great benefit for treating severe infections [68]. Oral amenamevir was recently approved in Japan to treat HZ and prevent post-herpetic neuralgia [169]. Both antivirals have not yet been studied in SOTRs.

### 8.2. Vaccines

Despite decades of vaccine development, the only HHV for which vaccines are approved is VZV. Life-attenuated varicella vaccines are included in the national vaccination programs of several countries worldwide to prevent varicella (chickenpox), but offer limited protection against HZ and are contraindicated in immunosuppressed individuals [170]. The live-attenuated HZ vaccine is effective in preventing HZ by approximately 70% in persons aged 50–59 years, but its effectiveness decreases with age and its use is also contraindicated in immunosuppressed individuals [171]. Contrastingly, the recombinant subunit vaccine, which is recently approved by FDA and EMA to prevent HZ, showed >95% protection for more than 5 years in healthy individuals in the same age group and is not contraindicated in immunocompromised individuals [172]. However, the vaccine’s high price may limit its general use in the aging population. The results of more large randomized trials and long-term studies are needed to evaluate its efficacy and safety in SOT practice [173,174].

Various types of vaccines were developed for HSV, HCMV, and EBV, but failed or still await the start of phase 3 trials. The format of these vaccines varies extensively, ranging from live-attenuated HHV, vector viruses encoding HHV proteins, HHV peptide mixes, HHV DNA, and more recently, mRNA vaccines. The phase 2 data of various vaccines are promising, showing good safety, variable immunogenicity, and even occasionally preventive efficacy [175,176,177]. Currently, the trial of a bivalent replication-deficient RNA vector vaccine to prevent HCMV infection in D+/R− living-donor kidney transplant candidates is ongoing (NCT03629080). A major obstacle in vaccine development is the knowledge of the correlates of protection, which evidently varies between the different HHVs. Because of the high species’ restriction of HHV, no appropriate animal models exist to determine the correlates of protection and test vaccine efficacy. Consequently, studies on well-stratified SOTRs are warranted to determine the recipient’s immune status before and after SOT, to identify HHV-specific protective immune parameters that will inform us which type of adaptive immunity to monitor and which viral proteins to include in future vaccines. Indeed, the identification of the recombinant subunit vaccine-induced correlates of protection against HZ is of major importance, especially for the development of a vaccine against its closely related herpesvirus HSV.

Besides developing new vaccines, another important challenge in the field of organ transplantation is how to improve vaccine responses in SOTRs. The COVID-19 pandemic made it clear that COVID-19 vaccine-induced immune responses were impaired in SOTR, which led to relatively high morbidity and mortality [178,179]. This resulted in studies investigating alternative vaccination strategies to improve vaccine-induced immune responses [180]. Some possible alternative strategies are administering booster doses, by means of intranasal or intradermal vaccine application, and temporarily adjusting immunosuppressive maintenance regimes.

### 8.3. Adoptive Immunotherapy

An alternative to classic antiviral therapy for SOTRs could be the use of adoptive immunotherapy. Examples are the transfusion of EBV-specific (cytotoxic) T-cells as a new approach to prevent PTLD and the use of HCMV-specific (cytotoxic) T-cells to prevent HCMV disease [117,181,182]. To prevent PTLD, the autologous or allogeneic (HLA-matched banked third-party donors) T-cells were used as a preemptive therapy, mainly in HSCT and in only a small amount of SOT patients, which were considered at high risk of PTLD based on the EBV load, serology, or type of transplant [183]. Virus-specific T-cells can also be used as a treatment. In 2019, the first report of the administration of expanded autologous HCMV-specific T-cells for refractory HCMV disease in SOTRs (KTR, LuTR, and HTR) showed that this strategy was safe and effective in 11 out of 13 participants [184]. In addition, the use of EBV-T-cells for rituximab-refractory EBV-associated lymphoma seems to be a promising potential therapy for SOTRs [185]. There are recent case reports of two kidney transplant recipients with primary central nervous system lymphoma successfully treated with Ibrutinib (first-generation Bruton’s tyrosine kinase inhibitor) and allogeneic EBV-specific T-cells, leading to an ongoing phase 1 trial (ACTRN12618001541291) [186].

Another type of allogeneic T-cell immunotherapy is Tabelecleucel, an investigational EBV-specific T-cell immunotherapy, which targets and eliminates EBV-infected cells. This treatment is currently assessed in a phase 3 trial for its efficacy and safety for the treatment of EBV-associated PTLD after the treatment failure of rituximab or rituximab plus chemotherapy in SOTRs (NCT03394365).

Other new developments in adoptive immunotherapy include the adoptive transfer of pamidronate-expanded gamma delta T-cells (Vγ9Vδ2 T-cells), chimeric antigen receptor (CAR)-T cells, and ex vivo engineered EBV-specific cytotoxic T cells that are tacrolimus-resistant [187]. CAR-T cells were initially developed to target CD19, a B-cell specific antigen, but can also be used to target viral membrane proteins including latency-associated membrane proteins and the gp350 of EBV [188,189]. However, the first report describing the use of CAR-T cells in SOTR with late-onset EBV-negative refractory PTLD showed no clinical response and several adverse effects [190].

Recently, a prophylactic infusion of donor-derived antigen-specific T-cells targeting seven different pathogens (i.e., HCMV, EBV, adenovirus, VZV, influenza virus, BK virus, and *Aspergillus fumigatus*) were given to 11 HSCT recipients. The generation of this multi-pathogen T-cell preparation was feasible and resulted in an increase in antigen-specific effector memory CD8 T-cells, but was associated with graft versus host disease [191]. The same strategy of using multiple-pathogen-directed T-cells was investigated in a phase 2 study using banked allogeneic virus-specific T-cells directed towards five different viral pathogens (i.e., EBV, adenovirus, HCMV, BK virus, and HHV6) in HSCT recipients with drug-resistant infections. Results to date indicate that this therapy may be safe and effective [192]. Whether adoptive immunotherapy in SOTRs will be of therapeutic value, without increasing rejection risk, remains to be investigated.

### 8.4. Viral Genome Editing

Whereas antivirals and vaccines are effective by inhibiting viral replication and to prevent overt HHV diseases, both strategies leave latent HHV intact, posing a lifelong risk of reactivation. Moreover, vaccination will never result in sterile immunity to HHV. The inactivation of latent HHV by viral genome editing is the ultimate therapy, but a major challenge to herpes virologists. Most HHV have their own anatomic/cellular site of latency (e.g., neurons and lymphocytes) and the HHV genomic site to destroy in order to prevent reactivation is largely unknown. The CRISPR/Cas9 system was applied in the genome editing and disruption of latent HCMV, EBV, HSV, and HHV8 infections in vitro [193,194]. Although it is a promising strategy to prevent and control HHV latency, it is too early to be considered in the clinic. Major drawbacks include specifically targeting the cells that contain latent HHV, genomic the site to target, and particularly safety issues like off-target editing with a potential destructive effect on the host.

## 9. Conclusions

Human herpesviruses are still dreaded pathogens for solid organ transplant recipients. Prevention and treatment largely depend on a limited number of antiviral drugs and registered vaccines are only available against VZV. However, the development of novel antiviral agents and vaccines are in full swing. Adoptive immunotherapy as well as methods to inactivate HHV latency are explored to limit the burden of and improve the prognosis of HHV-induced diseases.

## Data Availability

Data sharing not applicable.

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
