# Peer review of "Current Perspectives on the Management of Herpesvirus Infections in Solid Organ Transplant Recipients"

_viruses, 2023, doi:10.3390/v15071595_

Round 1

Reviewer 1 Report

In the manuscript entitled: “Current Perspectives in the Management of Herpesvirus Infections in Solid Organ Transplant Recipients” Malahe et al. give a thorough overview of the complications, prevention and treatment for all herpesvirus infections relevant in the context of the SOT. The review is very well organized and follows the order of relevance of particular herpesvirus to the SOT setting, starting with the most important HCMV. The : Incidence, Complications, Prevention and Treatment are discussed for each virus and this structure of the review allows for a very good overview of issues related to each virus at the same time if needed one can compare different aspects between the different infections. The final section of the manuscript entitled: “Future perspectives” includes an up to date information on Anivirals, Vaccines, Adoptive immunotherapy and even Viral genome editing as a potential future approach. All of these are discussed in the context of SOT, different approaches are mentioned, current clinical trials are listed and challenges are also clearly pointed out. In my opinion the authors presented an up to date information in a very well organized manner, which gives a good and easy to follow overview for the reader. I have only minor comments, which I list below.

1. It might be a good idea to add a short section to the introduction giving an overview of available a) prevention strategies and b) therapeutic strategies, including short definitions of eg. antiviral prophylaxis and preemptive therapy. I mean just general strategies that can be used.

2. In line 125, where letermovir is mentioned perhaps it would be a good idea to mention that it is an inhibitor of terminase complex. Of course, it is well known by now, but since it has a different mechanism of action then the other inhibitors, it might be worth stating what is that mechanism.

3. I believe that it would be beneficial to mention a number in the case of prevalence also for EBV, as it is done for other viruses.

Author Response

Responses to reviewer #1:

1) ‘’It might be a good idea to add a short section to the introduction giving an overview of available a) prevention strategies and b) therapeutic strategies, including short definitions of eg. antiviral prophylaxis and preemptive therapy. I mean just general strategies that can be used.’’

  • Response: We have added an overview of the current available prevention (including a short definition of antiviral prophylaxis and preemptive therapy) and therapeutic strategies to the introduction (page 2, lines 47-56).

2) ‘’In line 125, where letermovir is mentioned perhaps it would be a good idea to mention that it is an inhibitor of terminase complex. Of course, it is well known by now, but since it has a different mechanism of action then the other inhibitors, it might be worth stating what is that mechanism.’’

  • Response: We have added the requested info in page 4 (line 166).

3) ‘’I believe that it would be beneficial to mention a number in the case of prevalence also for EBV, as it is done for other viruses.’’

  • Response: We have added the requested info in page 6 (lines 268-269).

Reviewer 2 Report

This review is well written and very comprehensive regarding the role or incidence of human herpesvirus infections in solid organ transplant patients.  I have a couple of comments and suggestions.

1.  The discussion of EBV should include the topic that EBV viral load is often routinely monitored in recipient's blood as a predictor of PTLD.  Reaching the viral load threshold has been used to determine when to reduce immunosuppressive therapy in order to prevent PTLD onset.

2. Kaposi's sarcoma is well documented among SOT recipients.  It is assumed that this is due to reactivation from the donor.  There have been several papers published discussing this reactivation (much like the CMV discussion).  Given the lengthy discussion in CMV regarding donor vs recipient, this should also be explained for HHV-8.

3.  In the discussion, the authors state that adults would be positive for 3 of the 9 human herpesviruses.  Given the percentage of infection among adults I would argue its higher than that.  HSV-1 is over 80% seropositive among adults.  HHV-6 and HHV-7 are at essentially 100% by age of 3-4. CMV is over 60% in the US and EBV is around 90-95%.  VZV, among adults born prior to 1970, would also be over 50% (too old to have been given the vaccine).  Thus, current adults should have 5-6 of the known herpesvirus infections.

Author Response

Response to reviewer #2:

1) ‘’The discussion of EBV should include the topic that EBV viral load is often routinely monitored in recipient's blood as a predictor of PTLD. Reaching the viral load threshold has been used to determine when to reduce immunosuppressive therapy in order to prevent PTLD onset.’’

  • Response: We have added this info to the ‘’prevention’’ part of the EBV section (page 6, lines 290-293).

2) ‘’Kaposi's sarcoma is well documented among SOT recipients. It is assumed that this is due to reactivation from the donor. There have been several papers published discussing this reactivation (much like the CMV discussion). Given the lengthy discussion in CMV regarding donor vs recipient, this should also be explained for HHV-8.’’

  • Response: We have added additional data to the HHV8 section on the factors that can lead to HHV8 reactivation and on the different mechanisms of HHV8-induced diseases (page 8, lines 402-404 and 407-410).

3) ‘’In the discussion, the authors state that adults would be positive for 3 of the 9 human herpesviruses. Given the percentage of infection among adults I would argue its higher than that. HSV-1 is over 80% seropositive among adults. HHV-6 and HHV-7 are at essentially 100% by age of 3-4. CMV is over 60% in the US and EBV is around 90-95%. VZV, among adults born prior to 1970, would also be over 50% (too old to have been given the vaccine). Thus, current adults should have 5-6 of the known herpesvirus infections.’’

  • Response: We agree with the reviewer’s reasoning. We have changed the sentence ‘’the majority of adults are latently infected with at least 3 of 9 human herpesviruses’’ to ‘’the majority of adults are latently infected with at least 5 of 9 human herpesviruses’’ (page 10, line 470).

Reviewer 3 Report

In this review article entitled: Current Perspectives in the Management of Herpesvirus Infections in Solid Organ Transplant Recipients, Malahe SRK. et al. reviewed nine distinct human herpes virus (HHV) infections in solid organ transplantation recipients (SOTR), especially in kidney KTR, liver (LiTR), lung (LuTR), and heat transplant recipients (HTR). The review discussed complications, recent advances in preventive measures, and treatment options against each HHV.

This review discussed the major issues, prevention, and treatment options in SOTR with HHV infection or reactivation from latency, especially with HCMV, EBV, and VZV infections. However, the review needs more details on the incidence rate and severity of the infections in complications with each HHV. A table should be provided to summarize the disease, rate of HHV infection, or reactivation in each type of SOTR. A lengthy paragraph on prevention against HCMV should be further divided into subsections, such as nucleoside drug, and non-nucleoside drug, passive immunotherapy, to outline strategies to prevent HCMV infection in different SOTRs. A figure or chart that summarizes the pro and cons of current therapies will also make this review more effective.

Figure 1 should be revised to highlight all therapy modalities against HHV infection in SOTR. The adoptive immunotherapy should include humoral and cell-mediated immunity. The figure legend lacks a description of the illustration.

Author Response

Response to reviewer #3:

1) ‘’However, the review needs more details on the incidence rate and severity of the infections in complications with each HHV. A table should be provided to summarize the disease, rate of HHV infection, or reactivation in each type of SOTR.’’ 

  • Response: The wide variety in SOTR types discussed, the individual patient/group’s disease history and therapy withholds us to draft a table with solid numbers on the incidence rate and severity of the infections in complications with each HHV worldwide. Moreover, we have touched upon these items in more general terms in the original

manuscript.

2) ‘’A lengthy paragraph on prevention against HCMV should be further divided into subsections, such as nucleoside drug, and non-nucleoside drug, passive immunotherapy, to outline strategies to prevent HCMV infection in different SOTRs.’’

  • Response: We agree with the reviewer and changed the manuscript accordingly. See restructuring including new headings on pages 2 to 4 of the revised manuscript.

3) ‘’A figure or chart that summarizes the pro and cons of current therapies will also make this review more effective.’’

  • Response: Likewise our comment to the reviewer’s point #1, we cannot draft a schematically and scientifically solid figure on the pros and cons of current therapies in the different SOTR cohorts and HHV type combinations. This, due to the wide variety in SOTR types discussed, the individual patient/group’s disease history and therapies treated and applied worldwide, respectively. Moreover, we have touched upon these issues in more general terms in the original manuscript.

4) ‘’Figure 1 should be revised to highlight all therapy modalities against HHV infection in SOTR. The adoptive immunotherapy should include humoral and cell-mediated immunity. The figure legend lacks a description of the illustration.’’

  • Response: The purpose of Figure 1 was to provide a simplistic representation of current perspectives, rather than a representation of all therapy modalities against HHV infection. However, given the incompleteness in therapy modalities, we decided to remove this figure from the manuscript.
